# Integrated Metabolomics and Proteomics Dynamics of Serum Samples Reveals Dietary Zeolite Clinoptilolite Supplementation Restores Energy Balance in High Yielding Dairy Cows

**DOI:** 10.3390/metabo11120842

**Published:** 2021-12-05

**Authors:** Sudipa Maity, Ivana Rubić, Josipa Kuleš, Anita Horvatić, Dražen Đuričić, Marko Samardžija, Blanka Beer Ljubić, Romana Turk, Damjan Gračner, Nino Maćešić, Hrvoje Valpotić, Vladimir Mrljak

**Affiliations:** Faculty of Veterinary Medicine, University of Zagreb, Heinzelova 55, 10000 Zagreb, Croatia; sudipabiotech.iitr@gmail.com (S.M.); irubic@vef.unizg.hr (I.R.); jkules@vef.unizg.hr (J.K.); horvatic.ani@gmail.com (A.H.); drazen.djuricic@kc.t-com.hr (D.Đ.); smarko@vef.unizg.hr (M.S.); bljubic@vef.unizg.hr (B.B.L.); rturk@vef.unizg.hr (R.T.); dgracner@vef.unizg.hr (D.G.); ninomacesic@gmail.com (N.M.); hvalpotic@vef.unizg.hr (H.V.)

**Keywords:** negative energy balance, zeolite clinoptilolite, proteomics, metabolomics, cows, serum

## Abstract

Dairy cows can suffer from a negative energy balance (NEB) during their transition from the dry period to early lactation, which can increase the risk of postpartum diseases such as clinical ketosis, mastitis, and fatty liver. Zeolite clinoptilolite (CPL), due to its ion-exchange property, has often been used to treat NEB in animals. However, limited information is available on the dynamics of global metabolomics and proteomic profiles in serum that could provide a better understanding of the associated altered biological pathways in response to CPL. Thus, in the present study, a total 64 serum samples were collected from 8 control and 8 CPL-treated cows at different time points in the prepartum and postpartum stages. Labelled proteomics and untargeted metabolomics resulted in identification of 64 and 21 differentially expressed proteins and metabolites, respectively, which appear to play key roles in restoring energy balance (EB) after CPL supplementation. Joint pathway and interaction analysis revealed cross-talks among valproic acid, leucic acid, glycerol, fibronectin, and kinninogen-1, which could be responsible for restoring NEB. By using a global proteomics and metabolomics strategy, the present study concluded that CPL supplementation could lower NEB in just a few weeks, and explained the possible underlying pathways employed by CPL.

## 1. Introduction

High-yielding dairy cows often suffer from frequent NEB around calving time, when the energy demand for maintenance and lactation are not met by dietary energy intake [1,2]. A severe NEB during calving can result in an altered metabolic disorder and it might take several weeks or months for the cows to regain a positive energy balance status [3]. Cows compensate for this energy deficit by mobilizing body reserves, which consist mainly of body fat [4]. Maladaptation to NEB periods not only has a negative impact on milk production, but also increases the risk of postpartum diseases such as clinical ketosis, mastitis, and fatty liver, and may also result in infertility [5]. Recent studies indicate that shortening or omitting the dry period of dairy cows improves energy balance (EB) in early lactation [3,6,7].

Several earlier reports have focused on investigating indicators of NEB or metabolic status during the transition from the dry period to lactation [3,8,9]. Metabolic disturbances associated with NEB are characterized by high quantities of non-esterified fatty acids (NEFA) from adipose tissue and β-hydroxybutyrate (BHB). NEFA increase the production of ketone bodies, or their re-esterification into triglycerides (TG), causing fatty liver syndrome; BHB is associated with reduced milk production, increased risk of displaced abomasum, and hypocalcemia during NEB [10]. Other metabolic markers associated with NEB are increased concentrations of galactose-1-phosphate, acetyl-carnitine, glutamate, hippurate, phosphorylcholine, phosphocreatine, and several UDP-hexoses and decreased concentrations of acetate, acetoacetate, alanine, BHB, free carnitine, choline, and N-acetylated sugars found in both serum and milk [11]. Moreover, protein markers–especially acute phase inflammatory proteins such as complement 3 (C3), fibronectin (FN1), Plasma Kallikrein-sensitive glycoprotein (ITIH4), Alpha-2-HS-glycoprotein (AHSG), haptoglobin (HP), and ceruloplasmin (CP)–were previously reported to be higher in milk and mammary epithelial cells during NEB [1]. Other proteins that are related to cholesterol synthesis, such as apolipoprotein A1 (APOA1), sterol-4-alpha-carboxylate 3-dehydrogenase (NSDHL), and serpins such as endopin-2C and SERPIN domain-containing protein, were found to be downregulated in animals with severe NEB [1,12].

Zeolites are microporous hydrated aluminosilicates, with rigid anionic frameworks containing well-defined channels and cavities that contain exchangeable metal cations or neutral molecules that can be removed or replaced [13]. The unique microporous structure makes zeolites one of the most important inorganic cation exchangers that are important in different industrial applications, such as waste-water treatment, catalysis, nuclear waste, agriculture, animal feed additives, and other biochemical applications [13]. The widely tested zeolite suitable for medical applications is zeolite CPL, whose positive medical effect is mostly attributed to its basic clinoptilolite material properties, and in particular to reversible ion-exchange and adsorption capacity [14,15,16]. This central clinoptilolite characteristic, related to elimination of toxic agents, may be used to restore homeostasis and energy balance inside an animal’s body; for example, by eliminating the excessive production of ammonia and other gaseous products, including CO_2_ and H_2_S, which are due to imbalanced digestion or diverse pathogeneses [15,17]. In dairy cows, it has been shown that CPL supplementation improves rumen fermentation and leads to better absorption of nutrients during the digestion process [18]. In addition, the acidity of rumen content was also shown to decrease, utilization of nitrogen improves, and the quantity of unfavorable volatile fatty acids decreases [15].

Several reports over the years have investigated the different benefits and safety of using CPL to improve the health and energy status of animals [19,20,21]. However, limited information is available on the change of global metabolomics and proteomic profiles in serum that could provide a better understanding of the biological pathways and associated altered metabolic and proteomic status due to CPL supplementation during NEB. Thus, the present study aimed to achieve a deeper knowledge of the dynamics of the serum proteome and metabolome of the high-yielding breed, Holstein-Friesian, during their transition from pre- to post-parity in response to CPL-supplemented feeds. To our knowledge, this is the first study to use integrated global proteomics and metabolomics on serum samples to investigate the effects of CPL on NEB in cows during their transition from the dry period to lactation.

## 2. Results

### 2.1. Confirmation of NEB by NEFA and BHB Levels

An overview of the sample collection and the study workflow is shown in Figure 1. Following sample collection, BHB and NEFA concentration in the serum samples were recorded to confirm the status of NEB, which showed significant differences between the control and CPL-treated groups. The NEFA levels were recorded as significantly lower in all samples in groups 10pre-S, 5post-S, and 26post-S vs. 10pre-C, 5post-C, and 26post-C, respectively. However, no such difference was observed in 30pre-C vs. 30pre-S (Appendix A). In addition, BHB levels also showed significant differences, particularly between 5post-S vs. 5post-C (Appendix A). However, the recorded BHB levels did not show significant differences between other groups, which is at odds with the metabolomics data that showed a decrease in BHB in all samples collected after CPL supplementation. Details on BHB and NEFA levels are provided in Appendix A.

### 2.2. Metabolomics and Proteomics Statistical Analysis

The sparse partial least squares (sPLS-DA) method was employed to reveal group structure within and between groups. Although unsupervised PCA algorithms provide a means of achieving unbiased dimensionality reduction, sparse PLS-DA relies on the class membership of each observation and controls the number of components to produce robust and easy-to-interpret models in the form of 2D score plots between selected components (Figure 2). The Q-Q plots of the metabolite (features) and protein abundances were generated to assess whether the abundance distributions were comparable (Appendix A). For the plot of each comparison, the line represented a parametric curve with the points linearly distributed, suggesting normal distribution. Collectively, the global untargeted metabolomics profile resulted in the annotation and/or identification of a total of 5658 features (Appendix A), of which 62 metabolites were identified based on both retention time and mass to a standard. Prior to statistical analysis, all metabolite features (annotated and/or identified) showing intensities of less than 70% of the samples were filtered out. The *t*-tests between each comparison resulted in 523, 471, 427, and 502 features (Appendix A) for 30pre-S vs. 30pre-C, 10pre-S vs. 10pre-C, 5post-S vs. 5post-C, and 26post-S vs. 26post-C, respectively. Similarly, label-based proteomics led to the identification of 452 proteins, of which 301 were confidently identified with a minimum of two unique peptides (Appendix A). Based on 70% valid values and *t*-tests, 76, 64, 44, and 52 proteins were significantly different in abundance (Appendix A) for 30pre-S vs. 30pre-C, 10pre-S vs. 10pre-C, 5post-S vs. 5post-C, and 26post-S vs. 26post-C, respectively. The distribution of differentially expressed significant and non-significant metabolite features and proteins in each comparison was represented by volcano plots (Figure 3).

### 2.3. GO and KEGG Pathway Analysis

Proteomics data were used for GO annotation enrichment analysis to indicate over-represented subcategories in three groups such as biological processes, molecular function, and Reactome pathways (Figure 4A). This also provided a perspective of the change in distribution of the significant proteins in different FDR-enriched ontologies and pathways in each comparison. The annotations were available for 58, 24, 14, and 49 differentially abundant proteins in 30pre-S vs. 30pre-C, 10pre-S vs. 10pre-C, 5post-S vs. 5post-C, and 26post-S vs. 26post-C, respectively. The top five enriched subcategories in each GO showed overrepresentation and were found to be common in all the comparisons except Reactome pathways, such as the innate immune system, homeostasis, the regulation of insulin growth factor, and post-translational protein phosphorylation, which did not show overrepresentation in 5post-S vs. 5post-C. The most abundant subcategories were catalytic activity and protein-binding in molecular function with regulation of metabolic and biological processes and proteolysis in biological processes and complement cascade in Reactome pathways. The metabolomics data were used to indicate overrepresented subcategories in KEGG pathway enrichment (Figure 4B). Pathways such as propanoate, butanoate and glycerolipid metabolism, valine, leucine and isoleucine degradation, and citrate cycle were found to be the most abundant and common in all the group comparisons. Pathways such as glycosyl-phosphatidyl inositol-anchor metabolism, glycerophospholipid, arachidonic acid, and alpha-linoleic acid metabolism were exclusively enriched in 30pre-S vs. 30pre-C.

### 2.4. Hierarchical Clustering and Correlation Analysis

Hierarchical cluster analysis of metabolite features and protein abundance based on one-way ANOVA (q < 0.05) showed a marked difference between groups before and after CPL supplementation, resulting in the identification of 1091 and 64 significant metabolite features (Figure 5A) (Appendix A) and proteins (Figure 5B) (Appendix A), respectively. Correlation analysis helped in visualizing the overall correlations between different proteins. Protein candidates such as complement C3 (C3), inter-alpha-trypsin inhibitor (ITIH2), and haptoglobin, which were previously reported as influencing NEB by several studies, were chosen to show their correlation with each other and with the top correlated proteins based on their higher Pearson correlation values (r > 0.6), such as extracellular matrix protein 1 (ECM1), Ig-like domain containing protein, C4 binding protein (C4BPA), C3, fibronectin, (FN1), complement component 8 gamma chain (C8G), prothrombin (F2), complement component 1 (C1s), coagulation factor V (F5), thrombospondin (THBS1), carboxypeptidase N (CPN2), kinninogen 1 (KNG1), serotransferrin (TF), ITIH2, and β2-glycoprotein (Figure 5C). A list of the correlated proteins along with their statistical significance and fc is provided in Table 1. Similarly, in metabolomics data, 20 metabolite features (Table 2) were chosen based on their high Pearson correlation values with BHB to identify the positive and negative correlators of NEB (Figure 5D). Collectively, the top 10 positive and the top 10 negative metabolites were chosen to show their correlation with BHB. Of these, metabolites such as valproic acid, caprylic acid, and alpha hydroxyl lauric acid showed the highest positive correlation (r = 0.9) and methylnonenoate, succinic acid and capric acid showed the highest negative correlation (r = −0.9) with NEB.

### 2.5. Networking and Joint Pathway Analysis

Network analysis was performed on the correlated proteins and metabolite features to investigate their interaction and to understand whether they are functionally linked in different biological processes and pathways. Among all the correlated proteins, 12 proteins were functionally linked with protein-protein interaction (PPI) enrichment *p*-value of 1.0 × 10^16^, with 14 enrichment nodes and 31 enrichment edges suggesting significant interaction (Figure 6A). On the contrary, two proteins (ECM1 and HFE) that were correlated were not found in the network. Interconnected proteins such as HP, KNG1, THBS1, C1S, C4BPA, and C3 had a high interaction score of >0.8 and were found in most of the functional categories. Similarly, metabolite-metabolite interaction of the network explorer analysis module was used to investigate the interacting metabolite features. All of the correlated metabolites were observed to interact with each other by using external nodes such as oxygen, NADP, NADH, carbon dioxide, hydrogen sulfide, coenzyme A, and parathion in one network (Figure 6B).

Joint pathway analysis represented all matched FDR-enriched pathways based on the pathway enrichment method and pathway impact values (Figure 7). Pathways involving complement and coagulation cascade, glyoxylate and dicarboxylate metabolism, and ascorbate and alderate metabolism were observed to have the highest impact with the highest number of metabolites involved (>10). Furthermore, gene-metabolite interaction analysis was used to map all the correlated proteins and metabolite features, leading to the identification of one subnetwork with 11 nodes and 10 edges. This subnetwork consisted of proteins F2, FN1, C3, CP, SERPING1, and KNG1 and featured valproic acid, 4-hydroxy-2-oxoglutaric acid, leucic acid, butyric acid, and glycerol. The boxplots (Figure 8) represent the expression level of these nodes in different time points in control vs. CPL treatments. All proteins in the network showed a significant decrease in expression at all-time points with supplements. On the contrary, valproic acid and glycerol showed elevated expression at all-time points after supplementation, which is at the odds with the expression of BHB. In addition, expression of leucic acid is consistent with that of BHB; however, 4-Hydroxy-2-oxoglutaric acid showed higher abundances in all group comparisons except at the twenty-sixth day.

## 3. Discussion

In high-yielding dairy cows, the transition from gestation to intensive lactation is often marked by significant physiological changes that make the animals vulnerable to different diseases during the early lactation period [22]. This is also influenced by the lack of sufficient feed intake, which often does not supply the energy needed for maintaining basal metabolism and milk production, thereby causing NEB [2]. NEB leads to irreversible pathophysiological damage such as future infertility if not minimized during the transition period. Previous studies have suggested that CPL dietary supplementation may improve energy status and reproductive efficiency during the postpartum period in cows [23,24,25]. The effect of long-term use of CPL have also been investigated, on different physiological statuses such as improved rumen digestion, increased glucose level, reduced subclinical ketosis, and reduced oxidative stress [26]. In addition, the use of CPL has been evaluated extensively in the treatment of different tumors (e.g., skin and mammary gland tumors) in dogs and have been found to be effective in decreasing tumor size significantly without any adverse toxicological consequences [27]. Notwithstanding the wide use of CPL in veterinary medicine, limited information is available on the relationship between proteomic and metabolomics dynamics and CPL supplementation, to provide a better understanding of the complexities of NEB and help facilitate CPL-based therapeutic intervention in husbandry to improve animal health and milk quality. Thus, we undertook an integrated proteomics and metabolomics study of serum samples collected from prepartum (30 and 10 days) and postpartum (5 and 26 days) periods, in control and CPL-supplemented groups, to examine the possibility of identifying differentially expressed serum metabolites and proteins that are representative of the nutritive modulation of dietary CPL on NEB.

Proteins related to acute inflammatory and immune responses such as C3, FN1, prothrombin (F2), alpha-1B-glycoprotein (A1BG), beta-2-glycoprotein 1 (APOH), IGL@ protein (IGL), HP, and CP were higher in all four groups that did not receive CPL, as compared with the groups that received CPL. This is consistent with a previous report that also observed the overexpression of these proteins in milk during early lactation, indicating NEB [5,28,29,30]. In addition, our study showed the change in enrichment of the ontologies and pathways involving catalytic activity, protein binding, endopeptidase activity, proteolysis, regulation of metabolic processes, the innate immune system, and homeostasis during different phases of NEB, which reduced significantly during 10 days prepartum and 5 days postpartum (but were found to be highly enriched at 26 days postpartum). This could imply that although CPL could not prevent NEB, it restored the energy balance in under a month of parturition, which otherwise might have extended for up to 20 weeks [31]. Furthermore, in the networking analysis of the correlated proteins, candidates such as the C4b-binding protein (C4BPA), the inter-alpha-trypsin inhibitor (ITIH2), thrombospondin (THBS1), and the complement C1S subcomponent (C1S) were lower in all the groups with CPL treatment that were previously reported to be overexpressed in oxidative stress [32,33,34,35]. On the contrary, other inflammatory proteins such as carboxypeptidase N (CPN2), serotransferrin (TF), and coagulation factor V (F5) showed no differential abundance, at 5 days postpartum, between the control and the CPL-treated groups; however, these proteins were found to be lower at 26post-S vs. 26post-C. This suggests that although CPL treatment did not significantly influence the ongoing active inflammatory response 5 days postpartum, it restored the energy deficit with the possibility of the endometrium reaching a more advanced stage of repair 26 days postpartum.

The metabolomics data showed that the BHB concentration was significantly downregulated by at least two-fold after CPL supplementation at different time points; however, the results provided by the biochemical analyzer after sample collection were noted to differ, not being significantly downregulated except at 5 days postpartum. Nevertheless, the NEFA concentrations recorded by the analyzer were found to be significantly less in sample groups treated with CPL. Under the condition of CPL supplementation, the metabolites that were downregulated with the most positive Pearson correlation values (r > 0.9) with BHB were 9-oxo-nonanoic acid (r = 0.94), D-Leucic acid (r =0.91), (1S,2R,4S)-(-)-Bornyl acetate (r = 0.90), and 8-Methylnonenoate (r = 0.90), followed by moderately positive correlation values (0.9 < r > 0.7) with capric acid, botrydial, butabarbital, and oleamide. Lipid peroxidation is one of several oxidative processes that take place during NEB in cows [36]. Metabolites such as oxo-nonanoic acid and 8-methylnonenoate are involved in lipid peroxidation [37], and their downregulation after CPL supplementation in our study evidently suggested the restoration of EB. In addition, the expression of ketogenic compounds such as D-Leucic acid and capric acid were found to be low after CPL supplementation, which could possibly suggest lowered gluconeogenesis, which is characteristic of NEB in compensating for glucose shortage [38]. Metabolites such as oleamide, butabarbital, and botrydial, which were observed to be downregulated after CPL supplementation, were not previously associated with any direct mechanisms related to NEB; however, their elevated expression was observed in processes of cognitive malfunction [39,40], one of the several physiological processes underlying NEB [3]. On the contrary, the most negative correlators of BHB (r < −0.6) were glycerol (r = −0.85), cis-1,2-Cyclohexanediol (r = −0.76), valproic acid (r = −0.68), and Caprylic acid (r = −0.68), which is in line with previous reports that suggested upregulation of these metabolites in maintaining EB. For example, the metabolic pathway of glycerol was found to be much closer to that of glucose, when compared with other glucose precursors, and hence it is used in dairy diets during calving [41]. In addition, valproic acid and caprylic acid were both reported as increasing insulin sensitivity and improving energy storage [42,43]; nevertheless, no such effect was observed in cis-1,2-Cyclohexanediol.

Of the top 20 correlators of BHB, metabolites such as eugenol, heptanoic acid, isoeugenol, proline betaine, valproic acid, caprylic acid, 8-Methylnonenoate, oleamide, capric acid, 4-hydroxy2-oxoglutaric acid, and glycerol were observed to form a network with BHB. Interconnected decanoic acids such as valproic acid and caprylic acid were upregulated on the twenty-sixth day post-parturition after CPL supplementation, and capric acid was downregulated at alle time points except the twenty-sixth day post-parturition after CPL supplementation. All these polyunsaturated fatty acids were previously reported to be potent inhibitors of infections such as mastitis in cows and salmonella in chickens [44,45,46,47,48,49]. Although the upregulation of valproic acid and caprylic acid after CPL supplementation could be justified, the exact reason for the downregulation of capric acid is still not known. Similarly, both eugenol and isoeugenol were reported to have prooxidant activity that included allergic and inflammatory reactions [50] and were observed to be downregulated after CPL supplementation. Moreover, free oleic acids from oleamide, which were previously reported as having adverse effects on bovine granulosa cell function during NEB [51], were also observed to be downregulated after CPL treatment.

Joint pathway analysis revealed the complement and coagulation cascade pathway to be the most FDR-enriched among all the pathways investigated in our study. All complement components along with the plasminogen precursor, the factor XIIa inhibitor precursor, the P-selectin precursor, the alpha-1-antiproteinase precursor, prothrombin, alpha-2-antiplasmin, and the coagulation factor V involved in the cascade were downregulated in CPL-treated samples. Previous studies have indicated that overactivation of this system has associated factors in many pregnancy complications as the cascade deposits several split products on the cell membrane, ultimately creating a cytotoxic cell lysis complex, inflammation, and tissue injury including free circulating anaphylatoxins such as C3a and C5a [52]. This might suggest that CPL not only treated NEB but also reduced the associated reproductive complications. Following this pathway, glyoxylate and dicarboxylate metabolism, ascorbate metabolism, aldarate metabolism, and butanoate metabolism were among the most enriched ones. Metabolites commonly involved in these pathways were 4-hydroxy-2-oxoglutarate, 2-hydroxy-3-oxoadipate, and D-Glucurono-6,3-lactone that were upregulated, and benzene butanoic acid and D(-)-beta-hydroxy butyric acid that were downregulated under CPL treatment. All the pathways were glucogenic and increased the blood sugar level to reduce NEB in the postpartum stage. On the contrary, pathways such as nitrogen metabolism, lipoic acid metabolism, glutanthione metabolism, and taurine and hypotaurine metabolism were the least FDR-enriched pathways, indicating no such significant differential impact of these pathways after CPL treatment. Collectively, six proteins and five metabolites showed interaction, which included valproic acid, leucic acid, 4-hydroxy-2-oxoglutaric acid, F2, glycerol, SERPING1, KNG1, FN1, BHB, CP, and C3. Amino acids derived from muscle were important gluconeogenesis substrates in starvation that resulted from NEB. Previous studies reported the use of amino acids derived from muscles as gluconeogenesis substrates during starvation in NEB. Our report showing the downregulation of ketogenic amino acids such as leucic acid, and their converted product BHB might be a possible indication of inhibition of gluconeogenesis after CPL treatment [53,54]. This is in coherence with the upregulation of valproic acid in our study, which was also previously reported to inhibit gluconeogenesis [55,56].

## 4. Materials and Methods

### 4.1. Animals and Housing

The experimental protocol for the study was approved by the Ethical Committee of the Faculty of Veterinary Medicine, University of Zagreb, Croatia. The research protocol and animal management complied with Directive 2010/63/EU of the European Parliament (2010) on the protection of animals used for scientific purposes (Class: 640-01/14-17/51; Registry number: 251-61-01/139-14-1). In total, 16 high-yielding Holstein-Friesian dairy cows averaging 680 ± 30 kg of body weight, between 2–7 years of age, were chosen and randomly assigned to the control (*n* = 8) and CPL-treated groups (*n* = 8) All the cows were housed in a free-stall barn with straw bedding, fed a ration composed of haylage, corn silage, and a complete feed mixture with 19% of crude protein, and milked twice a day at 6 a.m. and 4 p.m. Before calving, during the drying-off period, and after calving during early lactation, the dietary forage-to-concentrate ratio was 75:25 and 60:40 respectively. Water was available ad libitum. All animals suffering from clinical mastitis, metritis, lameness, milk fever, abomasal displacement, retained placenta, or cystic ovarian dysfunction were excluded from the trial. The change in average body weight before and after parturition was recorded. The average weight of the 16 cows was 680 ± 30 kg at the start of the study (30 days before parturition). At the end of the study, the average body weight of the animals was 555.58 ± 21.22 kg.

### 4.2. Dietary Supplementation and Animal Group Assignment

To study the effects of CPL on NEB, 50gm of CPL was supplemented with feed. Zeolite CPL was modified by vibroactivation and micronization (Vibrosorb^®^, Viridisfarm, Podpićan, Croatia). Cows were randomly assigned to two groups: (i) the control group of non-treated cows (*n* = 8) and (ii) the treated group (*n* = 8) that received CPL in feed twice daily from the third month of gravidity to day 30 following parturition. Samples were collected from the control and treated groups on the following days: 30 days prepartum (control: 30pre-C, treatment: 30pre-S), 10 days prepartum (control: 10pre-C, treatment: 10pre-S), 5 days postpartum (control: 5post-C, treatment: 5post-S), and 26 days postpartum (control: 26post-C, treatment: 26post-S). Hence, samples were collected from four different time points from eight animals in each group (Figure 1). No significant differences in milk yield were found between the groups of cows (the CLP group was 8325.5 ± 628.8 kg vs. the control group which was 8050 ± 586.8 kg).

### 4.3. Serum Collection and Sampling Strategy

Blood samples were taken after the morning milking with the BD Vacutainer^®^ blood collection system (Becton, Dickinson and Company, Franklin Lakes, NJ, USA) from the tail vein (*v. coccygea*) into tubes without an anticoagulant but with a clot activator. After clotting at room temperature for 1 h, blood samples were centrifuged at 1500× *g* for 15 min. Sera were separated and stored at −70 °C until analysis. Serum BHB and NEFA concentrations were determined with the Beckman Coulter AU 640 automated biochemical analyzer (Beckman Coulter Biomedical Ltd., München, Germany) using commercially available reagent kits (Randox Laboratories Ltd., Crumlin, UK). Based on BHB and NEFA concentrations, collectively 64 serum samples were chosen for the metabolomics and proteomics experiments.

### 4.4. Sample Preparation for Metabolomics and Proteomics

For metabolite extraction, 25 µL of each serum aliquot were mixed with 1000 µL of extraction solvent (Chloroform: Methanol: Water, 1:3:1) to precipitate the proteins. A total of 10 µL of each sample (control and disease) was pooled, and a volume of 25 µL of pooled sample was also subjected to extraction. The matrix blank sample contained only extraction solvent. All samples (serum samples, pooled samples, and matrix blank) were subsequently vortexed and centrifuged at 13,000× *g* for 5 min at 4 °C. The supernatant (200 µL) was transferred to a screw-top vial and stored at −80 °C for liquid chromatography-mass spectrometry (LC-MS) analysis.

Relative quantification of proteins of the serum samples was performed using tandem mass tag (TMT)-based shotgun methods previously described [57]. In brief, total protein concentration was determined using BCA assay (Thermo Scientific, Rockford, IL, USA). An amount of 35 μg of total serum proteins from all individual samples and an internal standard (a pool of all samples) was diluted using 0.1 M triethyl ammonium bicarbonate (TEAB) (Thermo Scientific) and subsequently reduced (20 mM DTT, 60 min, 55 °C), alkylated (30 mM IAA, 30 min, room temperature in the dark), and acetone-precipitated (6 volumes, overnight,−20 °C). Protein pellets were collected by centrifugation, dissolved in 0.1 M TEAB, and digested using 1:35 *w*/*w* trypsin (Promega, Madison, WI, USA) at 37 °C overnight. Peptides were labeled with freshly prepared TMT sixplex reagents (Thermo Scientific). Differentially TMT-modified samples were combined with the internal standard into a new tube, aliquoted, dried, and stored at −20 °C for further analysis.

### 4.5. LC-MS Measurements

Both metabolomics and proteomics quantification were performed in a q exactive orbitrap mass spectrometer (Thermo Fisher Scientific, Bremen, Germany). For metabolomics, a SeQuant^®^ ZIC^®^-pHILIC column (150 mm × 4.6 mm, 5 μm column, polymer) (Merck-Millipore, Darmstadt, Germany) was used with gradient elution under alkaline conditions in a UHPLC system (Thermo Fisher Scientific, Germering, Germany). Mobile phase A was 20 mM ammonium carbonate in water and mobile phase B was 100% acetonitrile. The following gradient was used at a flow rate of 0.300 mL/min and 25 °C: 0.0–15.0 min 80–20% B, 15.0–17.0 min decrease to 5% B and at 17.0 −27.0 min switched to 80% B. The injection volume was 10 μL and samples were maintained at 5 °C prior to injection. The MS was operated in polarity-switching mode and the MS settings had a resolution of 70,000, AGC of 1e6, *m/z* range of 70–1050, sheath gas 40, auxiliary gas 5, sweep gas 1, and capillary temperature of 320 °C. For positive mode ionization the source voltage was +3.8 kV, S-Lens RF Level 60.00; for negative mode ionization the source voltage was −3.8 kV.

For the proteomics experiment, peptides were loaded on the trap column (C18PepMap100, 5 mm, 100 A, 300 mm × 5 mm) (Thermo Fisher Scientific) and desalted for 12 min at the flow rate of 15 uL/min using the ultimate 3000 system with an autosampler (Thermo Fisher Scientific). Subsequently, TMT-labelled peptides were then separated on the analytical column (PepMap™ RSLC C18, 50 cm × 75 μm) (Thermo Fisher Scientific) using linear gradient 5–55% mobile phase B (0.1% formic acid in 80% ACN) over 120 min at the flow rate of 300 nL/min. Mobile phase A consisted of 0.1% formic acid in water. Ionization was achieved using the Nanospray Flex ion source (Thermo Fisher Scientific) containing a 10 μm-inner diameter SilicaTip emitter (New Objective, Littleton, MA, USA). MS was operating in positive ion mode using DDA Top8 method with parameters set as follows: full scan MS spectra range from *m*/*z* 350.0 to *m*/*z* 1800.0, resolution of 70,000, injection time 120 ms, AGC target 1 × 10^6^, isolation window ± 2.0 Da, and the dynamic exclusion 30 s. For HCD fragmentation, resolution was set to 17,500 and AGC target to 2 × 10^5^.

### 4.6. Database Searching

For metabolomics, MzXML files were uploaded in the Polyomics integrated Metabolomics Pipeline (PiMP) [58] and processed in both a positive ionization mode and a negative ionization mode version of each file. Metabolites were identified and annotated by KEGG and HMDB databases. Metabolites are referred to as identified if they are matched both by retention time and mass to a standard, while they are referred to as annotated if they are assigned putatively on the basis of mass. For proteomics, all LC-MS/MS data were searched using the SEQUEST algorithm implemented into Proteome Discoverer 2.3 (Thermo Fisher Scientific) against *Bos taurus* NCBInr FASTA files (downloaded 4 April 2019, 69,663 entries) to obtain peptide and protein identifications. For all searches, trypsin was specified as the enzyme for protein cleavage, allowing up to two missed cleavages. Carbamidomethyl (C) was set as a fixed peptide modification, while oxidation (M) and TMT sixplex (K, peptide N-terminus) were set as dynamic modifications. The precursor and fragment mass tolerances were set at 10 ppm and 0.5 Da, respectively. Both the peptide spectrum match and the protein false discovery rate (FDR) were set to 0.01 and determined using a Percolator node. The MS proteomics data were deposited in the ProteomeXchange Consortium via the PRIDE partner repository with the dataset identifier PXD012289. Metabolomics data were deposited in the EMBL-EBI MetaboLights database (DOI: 10.1093/nar/gkz1019; PMID:31691833) with the identifier MTBLS3423.

### 4.7. Statistical Analysis

All analyses of proteomics and metabolomics were performed in RStudio (version 1.4) unless otherwise mentioned. All proteins and metabolite features present in <70% of the samples were filtered out before proceeding, for statistical validation. The principal component analysis (PCA) plots for proteomics and metabolomics were generated using Perseus (Version 4.1.3) [59] and MetaboAnalyst (Version 5.0) [60]. Abundances of proteins and metabolite features were log-transformed and normalized based on the median values. Log-transformed abundances were normalized into z-scores to generate quantile-quantile plots (Q-Q plots). For both proteomics and metabolomics, parametric tests were employed as the collection of null *p* values were found to be uniformly distributed. For two-group tests, t-testing, and multiple group testing, one-way ANOVA was performed with FDR-adjustment by the Benjamini-Hochberg (BH) method. Metabolite features were deemed confident based on q < 0.05 and fold change −1 ≥ log fc ≥ 1. Proteins were deemed confident based on q < 0.05 and fold change log −0.2 ≥ log fc ≥ 0.2 (i.e., an average fc ≥ 20%, approximately two standard deviations (SD) from the mean of the calculated ratios, were considered significantly different). Volcano plots were generated for the distribution of all metabolite features, and proteins based on their statistical significance (q < 0.05) and fc heatmaps were generated on one-way ANOVA significant metabolites using the “Ward.D” hierarchical clustering method and distance measurement by the “canberra” method. Gene ontology (GO) and Reactome Pathway using the PANTHER Classification System [61] were performed on proteins, and KEGG pathway analysis using MetaboAnalyst [60] was performed on metabolite features that were significant by both FDR-adjusted *p* values and fc. Some notable protein biomarkers of NEB as described in previous reports were sorted and selected based on their fc and FDR-adjusted *p* values for generating a correlation plot. Similarly, a correlation plot was generated for the top 20 correlated metabolite features with BHB. Furthermore, networking analysis for the correlated metabolites and proteins were performed using MetaboAnalyst and String Application in Cytoscape (Version 3.8.0), respectively. In addition, joint pathway analysis was performed to investigate the FDR-enriched pathways involving the significant proteins and correlated metabolite features interacting collectively based on hypergeometric testing and topology analysis using the out-degree centrality method. Boxplots were generated to represent the change in abundance of the interacting metabolite features and proteins during prepartum and postpartum before and after CPL treatment.

## 5. Conclusions

In summary, this is the first study to use integrated global proteomics and metabolomics on serum samples to investigate the effects of CPL on NEB in cows during their transition from the dry period to lactation. We collectively identified and quantified 64 proteins using TMT-labelled quantitation, with a significant cohort appearing to play key roles in restoring EB after CPL supplementation. The results of metabolomics helped in identifying 83 metabolites based on both chromatogram and MS peaks, of which 21 were further chosen, based on their statistical significance and correlation with BHB, to understand their role in NEB and their alterations after CPL treatment. The results of proteomic and metabolomics alterations were consistent with previous studies and, in addition, expanded the number of proteins and metabolites identified from cow serum to date. Joint pathway and interaction analysis of differentially expressed proteins and metabolites revealed crosstalk among potential candidates such as valproic acid, leucic acid, 4-hydroxy-2-oxoglutaric acid, F2, glycerol, SERPING1, KNG1, FN1, BHB, CP, and C3, which could be responsible for restoring associated complication of NEB such as infertility, or infection such as mastitis. The results of the present study provide a foundation for better understanding of the effect of CPL on proteome and metabolome of the serum, which will ultimately help facilitate its use as an early therapeutic and husbandry-based intervention to improve energy status in cows.

## Figures and Tables

**Figure 1 metabolites-11-00842-f001:**
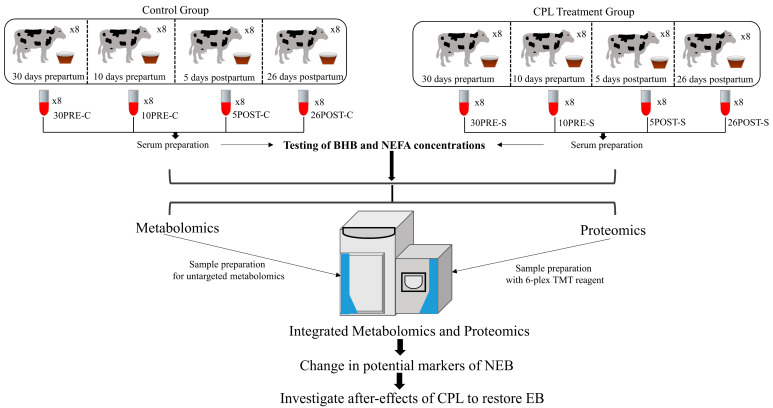
Study workflow. An overview of the procedure used for the identification and quantification of metabolites and proteins in control and CPL-treated groups at 30 and 10 days prepartum and 5 and 26 days postpartum.

**Figure 2 metabolites-11-00842-f002:**
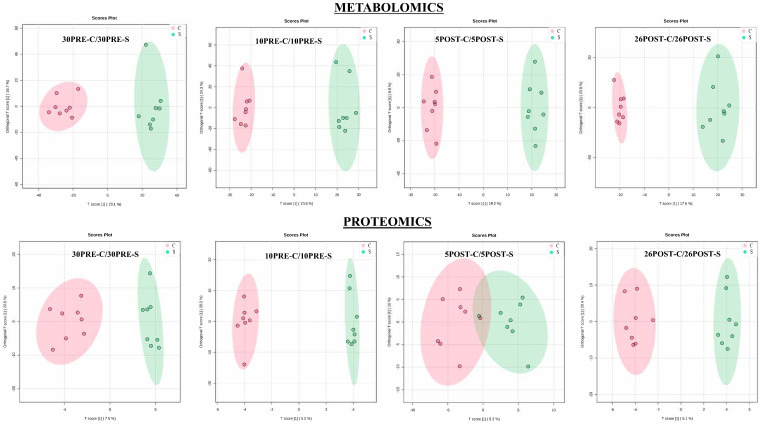
The combined data of serum metabolomics and proteomic profiles were separated in a sparse partial least squares discriminant analysis in four different comparisons.

**Figure 3 metabolites-11-00842-f003:**
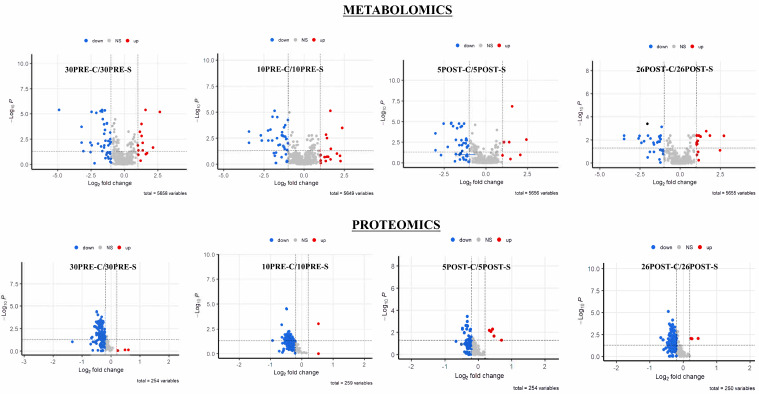
Distribution of differential expression of metabolites and proteins across group comparisons. Volcano plot showing the distribution of significant metabolite features and proteins based on t-tests versus fold change in 30pre-S vs. 30pre-C, 10pre-S vs. 10pre-C, 5post-S vs. 5post-C, and 26post-S vs. 26post-C.

**Figure 4 metabolites-11-00842-f004:**
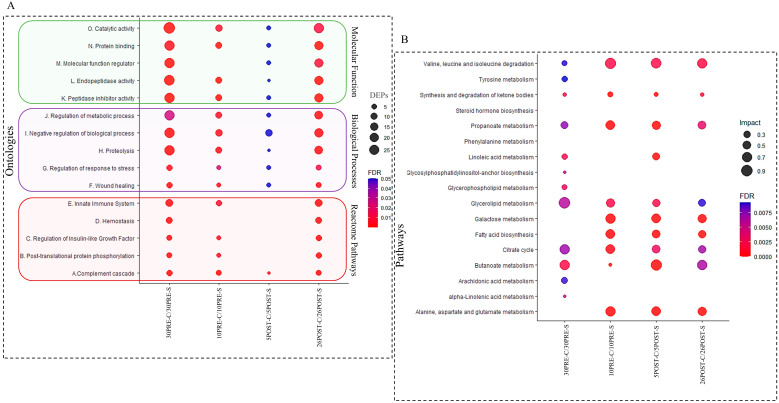
Gene ontologies and pathway enrichment for significant proteins and metabolite features. (**A**) The y-axis indicates the gene ontologies and Reactome pathway enrichment for proteins, and the *x*-axis indicates the group comparisons. The bubble size indicates the number of differentially expressed proteins (DEPs) involved in the ontologies. The color bar indicates the corrected *p*-value:e blue represents higher value and red represents lower value. (**B**) The y-axis indicates the KEGG pathway enrichment for metabolite features, and the *x*-axis indicates the group comparisons. The bubble size indicates the impact of the pathway represented by the number of differentially expressed metabolite features involved. The color bar indicates the corrected *p*-value: blue represents higher value and red represents lower value.

**Figure 5 metabolites-11-00842-f005:**
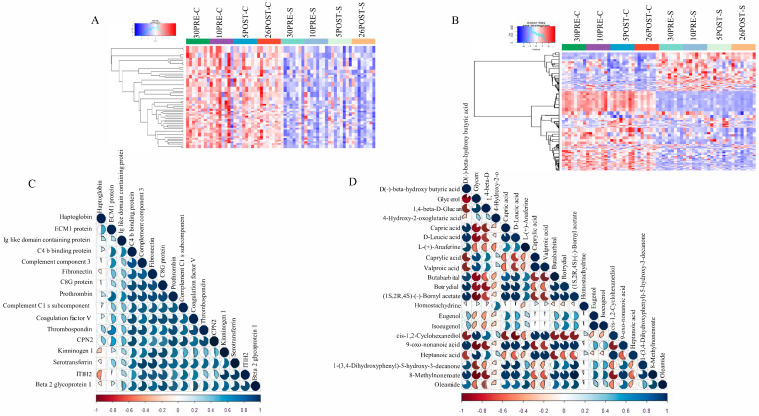
Hierarchical clustering and correlation plots. Intensities for proteins (**A**) and metabolite features (**B**) were z-score-transformed and are displayed as colors ranging from blue to red as shown in the key. The histogram in the key denotes that the distribution of the maximum z scores lies between −1 and +1. Both rows and columns are clustered using the “Ward.D” hierarchical clustering method and “canberra” distance. (**C**) Correlation analysis was performed using Pearson correlation values among 16 proteins with significant functional roles in NEB. (**D**) Correlation analysis was performed using Pearson correlation values of the top 20 metabolite features with BHB.

**Figure 6 metabolites-11-00842-f006:**
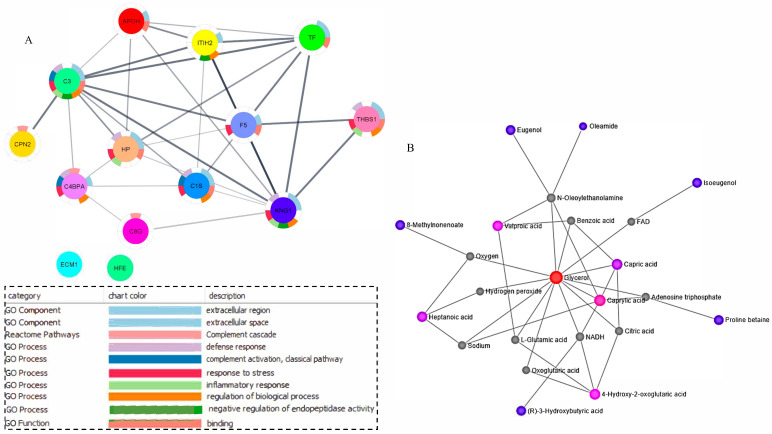
Networking analysis of the correlated proteins and metabolite features. (**A**) Networking analysis showed 12 correlated proteins to be interconnected with FDR-enriched ontologies. (**B**) Networking analysis showed 12 of 20 correlated metabolites, as well as BHB, to be interconnected. Other joining metabolites found in the network that were not found in our study were colored grey.

**Figure 7 metabolites-11-00842-f007:**
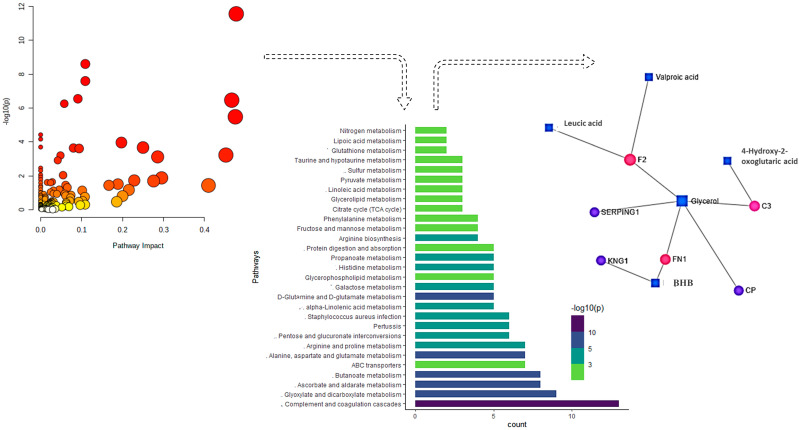
Joint pathway analysis represented all matched FDR-enriched pathways based on the pathway enrichment method and pathway impact values. Furthermore, gene-metabolite interaction analysis was used to map all the correlated proteins and metabolites, leading to the identification of one subnetwork with 11 nodes and 10 edges.

**Figure 8 metabolites-11-00842-f008:**
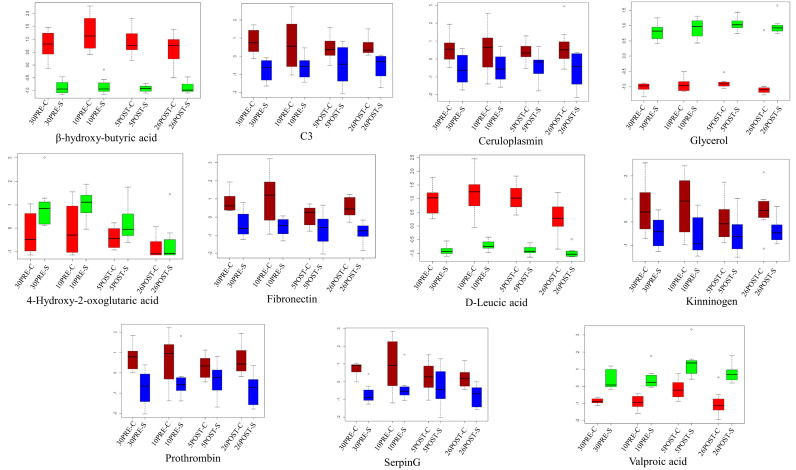
The intensities of the 11 nodes found in the joint pathway analysis were converted to z-scores to represent the distribution of their intensities in different groups as boxplots. The metabolites were represented by red and green for control and CPL-treated groups, respectively, and the proteins were represented by blue and dark red for control and CPL-treated groups, respectively.

**Table 1 metabolites-11-00842-t001:** List of correlated proteins with significance and fold change.

Acc Id	Gene	Protein	Log 30pre-C/S	Log 10pre-C/S	Log 5post-C/S	Log 26post-C/S	Q
Q28178	THBS1	Thrombospondin-1	−0.28	−0.51	−0.21	−0.37	0.001181
Q2UVX4	C3	Complement C3	−0.36	−0.30	−0.22	−0.17	0.001302
P00735	F2	Prothrombin	−0.33	−0.34	−0.13	−0.27	0.00257
Q28065	C4BPA	C4b-binding protein alpha chain	−0.41	−0.43	−0.25	−0.35	0.003106
A5D7R6	ITIH2	ITIH2 protein	−0.29	−0.37	−0.23	−0.38	0.003999
A5PJT7	ECM1	ECM1 protein	−0.52	−0.39	−0.19	−0.42	0.004807
G3X6K8	HP	Haptoglobin	−1.34	−0.73	−0.06	−0.02	0.006941
Q29443	TF	Serotransferrin	−0.39	−0.12	−0.19	−0.25	0.007474
G5E604	HFE	Ig-like domain-containing protein	0.78	0.14	0.38	−0.18	0.007474
Q0VCX1	C1S	Complement C1s subcomponent	−0.40	−0.24	−0.25	−0.42	0.009913
A8YXZ2	C8G	C8G protein	−0.21	−0.29	−0.21	−0.22	0.019858
P17690	APOH	Beta-2-glycoprotein 1	−0.22	−0.34	−0.25	−0.25	0.019858
Q28107	F5	Coagulation factor V	−0.31	−0.35	−0.16	−0.29	0.024159
A6QP30	CPN2	CPN2 protein	−0.15	−0.24	−0.18	−0.36	0.02606
P01044	KNG1	Kininogen-1	−0.25	−0.56	−0.18	−0.29	0.03451

**Table 2 metabolites-11-00842-t002:** List of correlated metabolite features in NEB with log-fc and q values.

Metabolites	KEGG	Log 30pre-C/S	Log 10pre-C/S	Log 5post-C/S	Log 26post-C/S	Q
(-)-Bornyl acetate	NA	−1.46	−1.36	−1.44	−1.42	0.000308
(S)-Homostachydrine	C08283	0.84	0.98	−0.42	0.98	0.018369
1,4-beta-D-Glucan	C00760	1.28	0.39	0.70	0.99	1.26 × 10^-8^
4-Hydroxy-2-oxoglutaric acid	C01127	1.60	1.40	0.65	4.09	0.000308
9-oxo-nonanoic acid	C16322	−1.32	−1.18	−1.15	−0.64	0.000957
Butabarbital	C07827	−1.06	−0.92	−0.65	−1.23	4.92 × 10^-13^
Capric acid	C01571	−1.68	−1.34	−1.30	−0.16	0.011789
Caprylic acid	C06423	0.65	0.77	0.74	1.59	5.24 × 10^-11^
D(-)-beta-hydroxy butyric acid	C01089	−3.20	−3.43	−3.03	−2.07	0.04743
D-Leucic acid	C03264	−1.52	−1.39	−1.51	−1.10	0.011789
Eugenol	C10453	−1.62	−1.84	−0.56	−1.06	2.20 × 10^-6^
Glycerol	C00116	1.57	1.63	1.58	1.91	0.000339
Heptanoic acid	C17714	0.53	0.83	0.67	1.73	9.61 × 10^-8^
Isoeugenol	C10469	−1.62	−1.84	−0.56	−1.06	2.20 × 10^-6^
L-(+)-Anaferine	C06183	−0.78	−1.20	−0.54	−0.39	0.000175
L-Aspartate	C00049	−0.13	−0.52	−0.24	−0.10	0.035791
Methylmalonate	C02170	−0.68	−0.44	−0.81	−0.40	2.09 × 10^-12^
Oleamide	C19670	−1.21	−0.67	−1.13	−0.50	0.00019
Valproic acid	C07185	0.65	0.77	0.74	1.59	5.24 × 10^-11^
Botrydial	C09622	0.00	0.00	−1.43	−1.70	7.21 × 10^-15^
cis-1,2-Cyclohexanediol	C12313	1.15	1.36	1.41	2.32	1.31 × 10^-14^
1-(3,4-Dihydroxyphenyl)-5-hydroxy-3-decanone	C17748	−0.75	−1.12	−0.76	−0.09	1.02× 10^-5^

## Data Availability

The mass spectrometry proteomics data have been deposited in the ProteomeXchange Consortium via the PRIDE [1] partner repository with the dataset identifier PXD027551.

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
