# Peer review of "Integrated Metabolomics and Proteomics Dynamics of Serum Samples Reveals Dietary Zeolite Clinoptilolite Supplementation Restores Energy Balance in High Yielding Dairy Cows"

_metabolites, 2021, doi:10.3390/metabo11120842_

Round 1

Reviewer 1 Report

  1. This is a novel study using metabolomics and proteomics to study the effect of zeolite clinoptilolite on the negative energy balance of dairy cows.
  2. Where is Table S1? Also the BHB and NEFA concentration results if possible BW changes to verify NEB status should be presented in the context.
  3. Have feed intake and BW change recorded? It is not very convincing to conclude the restoration of energy balance only based on BHB and NEFA concentrations.

Reviewer 2 Report

The manuscript explored the effects of supplementation of Zeolite clinoptilolite on serum metabolome and proteome profiles in peripartal dairy cows. This research article would expand our awareness on the effectiveness of adding mycotoxin binders at the systems biology level in dairy cattle. The manuscript is well-written and the methods are well-described. Please, find my comments below

INTRRODUCTION”

  • Change “can result to altered ..” into “can result in altered ..”
  • Change “disorder and may take ..” into “disorder and might take ..”
  • Cite a reference “..also result in infertility.”
  • Change “..have focused in investigating..” into “..have focused on investigating..”
  • Define BHB in “..adipose tissue and BHB.”
  • Replace “,beta-hydroxybutyrate, “ with “,BHB,”
  • Cite a reference in “..mammary epithelial cells during NEB.”
  • Cite a reference in “..other biochemical applications.”
  • Cite a reference in “..or in diverse pathogeneses.”
  • Cite a reference in “..nutrients during the digestion process.”
  • Cite a reference in “..energy status of the animals.”

MATERIALS AND METHODS

  • Add protocol # after “..University of Zagreb, Croatia.”
  • How many heifers and cows in each group?
  •  
